# Inherited Disorders of Coenzyme A Biosynthesis: Models, Mechanisms, and Treatments

**DOI:** 10.3390/ijms24065951

**Published:** 2023-03-21

**Authors:** Chiara Cavestro, Daria Diodato, Valeria Tiranti, Ivano Di Meo

**Affiliations:** 1Unit of Medical Genetics and Neurogenetics, Fondazione IRCCS Istituto Neurologico Carlo Besta, 20126 Milan, Italy; 2Unit of Muscular and Neurodegenerative Disorders, Ospedale Pediatrico Bambino Gesù, 00165 Rome, Italy

**Keywords:** coenzyme A, PANK2, PPCS, PPCDC, COASY, neurodegeneration, NBIA, cardiomyopathy, iron accumulation

## Abstract

Coenzyme A (CoA) is a vital and ubiquitous cofactor required in a vast number of enzymatic reactions and cellular processes. To date, four rare human inborn errors of CoA biosynthesis have been described. These disorders have distinct symptoms, although all stem from variants in genes that encode enzymes involved in the same metabolic process. The first and last enzymes catalyzing the CoA biosynthetic pathway are associated with two neurological conditions, namely pantothenate kinase-associated neurodegeneration (PKAN) and COASY protein-associated neurodegeneration (CoPAN), which belong to the heterogeneous group of neurodegenerations with brain iron accumulation (NBIA), while the second and third enzymes are linked to a rapidly fatal dilated cardiomyopathy. There is still limited information about the pathogenesis of these diseases, and the knowledge gaps need to be resolved in order to develop potential therapeutic approaches. This review aims to provide a summary of CoA metabolism and functions, and a comprehensive overview of what is currently known about disorders associated with its biosynthesis, including available preclinical models, proposed pathomechanisms, and potential therapeutic approaches.

## 1. Introduction

Coenzyme A (CoA) is an essential, widely distributed cofactor that is crucial to cellular metabolism, playing a central role in the production and breakdown of all significant sources of energy in the body [1]. Its importance has recently been highlighted by the identification of four human pathologies linked to inborn errors of metabolism (IEM) in the CoA cellular de novo biosynthetic pathway. Although they are all caused by sequence variants in genes coding for enzymes involved in the same pathway, these four clinical conditions can be divided into two groups, as they are characterized by very distinct clinical symptoms.

Variants in the enzymes catalyzing the first (pantothenate kinase type 2, PANK2) and last (CoA synthase, COASY) reactions of the pathway are responsible for the neurological conditions called pantothenate kinase-associated neurodegeneration (PKAN) [2] and COASY protein-associated neurodegeneration (CoPAN) [3], respectively. These two diseases belong to a highly heterogeneous group of rare neurodegenerative diseases known as neurodegenerations with brain iron accumulation (NBIA), and they share some typical clinical features, including movement disorders, cognitive impairment, and iron accumulation in basal ganglia [4,5].

A few years ago, a rapidly fatal dilated cardiomyopathy was linked to pathogenic variants in the second enzyme of the pathway (phosphopantothenoylcysteine synthetase, PPCS) with the discovery of three affected families [6,7]. Very recently, variants in the third enzyme of the pathway (phosphopantothenoylcysteine decarboxylase, PPCDC) were associated with a similar cardiac condition in two sisters [8].

At present, little is known about these pathologies, especially those that have only recently been discovered. Mitochondria and peroxisomes are the two major subcellular storage sites for CoA and acyl-CoAs, and the investigations about the pathogenesis of the diseases have primarily involved these two compartments [9]. Although mitochondrial dysfunction may play a key role in the two NBIA disorders, there is still little information on how its functionality affects the pathogenesis of the two cardiomyopathies.

While some possible mechanisms linking CoA biosynthesis to the pathogenesis of NBIA have been proposed, none of them has yet been widely accepted. Furthermore, it is not clear how variants in enzymes belonging to the same metabolic pathway lead to such different phenotypic manifestations. 

Here, we highlight what is known about these disorders, from the available preclinical models to the potential therapeutic approaches. Additionally, we aim to draw attention to the knowledge gaps to encourage their resolution, which could eventually result in the description of the proposed pathomechanisms.

## 2. Coenzyme A Homeostasis and Functions

CoA is a ubiquitous and crucial metabolic cofactor present in all living organisms, whose primary function is to act as an acyl group carrier and carbonyl-activating group in a plethora of metabolic reactions, regulating fundamental cellular processes, including cell growth and death, energy production, autophagy, signal transduction, protein modification, epigenetics, and the regulation of gene expression [10]. The molecule consists a central scaffold of 4-phosphopantothenic acid linked to an adenosine 3’,5’-diphosphate, and a β-mercaptoethylamine [11].

The de novo biochemical pathway by which the CoA is synthesized is remarkably conserved. The substrate for this process is vitamin B5 or pantothenic acid (Pan) (Figure 1A) [12].

In mammals, Pan mainly derives from the diet and the gut microbiota metabolism, and it is absorbed from the bloodstream through the sodium-dependent multivitamin transporter (SMVT). Once in the cell, pantothenate kinase (PANK) catalyzes the phosphorylation of Pan to 4′-phosphopantothenate (PPan) as the first step in the synthesis of CoA. The PANK enzyme exists in four isoforms (PANK1-4), encoded by four different genes, which differ in tissue expression and intracellular localization. PANK1a and PANK1b, coded by the same gene and differing only in their first exons, are both strongly expressed in the liver and kidney, followed by heart and skeletal muscle, with a cellular localization at nuclear and cytoplasmic levels, respectively [13]. PANK2 and PANK3 are both ubiquitously expressed, with higher levels in the liver and brain [14]. While PANK3 is localized in the cytosol, PANK2 is the only human pantothenate kinase mainly associated with mitochondria, with a specific localization in the intermembrane space. However, a nuclear fraction of the PANK2 enzyme has been identified [13]. PANK4 has been reported to encode a pseudo-enzyme lacking kinase activity but with an active phosphatase domain [15], which has been recently reported to be involved in cellular signaling as a substrate of AKT kinase [16]. Although to varied degrees of efficiency, all PANK1-3 forms are involved in kinase activity, offering a redundant function that can at least partially compensate for defects in one of them [17]. 

In the second step, the Phosphopantothenoylcysteine synthetase (PPCS) combines PPan and cysteine to form 4′-phospho-N-pantothenoylcysteine (PPan-Cys). Phosphopantothenoylcysteine decarboxylase (PPCDC) is responsible for the subsequent decarboxylation step, which results in the formation of 4′-phosphopantetheine (PPanSH). Finally, the bi-functional enzyme CoA synthase (COASY), possessing phosphopantetheine adenylyl-transferase (PPAT) and dephospho-CoA kinase (DPCK) domains, catalyzes the final two steps of the pathway with a reaction of adenylation to form dephospho-CoA (dPCoA), which is finally phosphorylated to CoA [12].

The crucial role that CoA plays in cellular homeostasis suggests that there is a carefully orchestrated intra and extracellular regulation with a defined balance between synthesis and degradation [18,19]. The PANKs-catalyzed reaction is the main rate-limiting step and controls the entire synthesis process via feedback inhibition of the enzyme by CoA and acyl-CoA [20]. The pathway’s last step is also susceptible to control since mammalian COASY activity is suppressed by feedback from CoA and CoA thioesters [18]. The total cellular CoA content is also regulated by the CoA degradation and salvage processes, which involve many enzymes and lead to the synthesis of the intermediates dPCoA, PPanSH, and pantetheine (PanSH) (Figure 1A). Nudix (nucleoside diphosphate-linked moiety X)-type motif (NUDT) hydrolyzes CoA and acyl-CoAs to PPanSH or its acylated derivative in peroxisomes and mitochondria [21]. Otherwise, CoA can be dephosphorylated to dPCoA in a process catalyzed by lysosomal alkaline phosphatases (AP) and then hydrolyzed to PPanSH by enzymes belonging to the ectonucleotide pyrophosphatase/phosphodiesterase (ENPP) family [22]. dPCoA can also be transformed into PanSH, which can return to PPanSH through phosphorylation activity by PANKs or by a still unknown kinase. PPanSH produced by CoA degradation could be recycled and come back into the pathway as a substrate of the COASY enzyme [19,21]. Moreover, PPanSH can in turn be dephosphorylated by PANK4 to form PanSH, suppressing CoA synthesis and regulating the abundance of acetyl-CoA and other acyl-CoAs, as well as CoA-dependent processes, such as lipid metabolism and cell proliferation [16]. Extracellularly, PanSH can be further hydrolyzed by vanins to cysteamine and Pan, which can in turn re-enter the cell as the substrate of CoA biosynthesis [19]. 

In addition to Pan, extracellular PPanSH has recently been hypothesized as a substrate for the synthesis of intracellular CoA [22]. PPanSH can enter the cells and be converted to CoA by COASY activities [23]. This alternative pathway avoids the first three enzymes and can be exploited to restore their impaired functionality. Studies using *Drosophila melanogaster*, *Caenorhabditis elegans*, and mammalian cells lacking PANK2 have shown that PPanSH can correct CoA deficiency [22]. PPanSH can come from a variety of sources: in addition to CoA degradation in the interstitial spaces, it can come from the metabolism of the gut microbiota, be ingested from food, and serve as a source of CoA precursors. The enzymes for CoA biosynthesis have a specific localization in the cellular compartments (Figure 1B) [13,24,25,26]. While PANKs enzymes localize to mitochondria or at the cytoplasmic level, where they can act as metabolic sensors sensitive to feedback inhibition by CoA and its thioesters [14], the subsequent two enzymes of the pathway, PPCS, and PPCDC, localize in the cytosol [21,26] and until now a single isoform for each of them was identified. 

Regarding the COASY enzyme, free cytoplasmic and nucleic localization have been observed in humans, although its primary localization was identified in the mitochondrial matrix or attached to the outer mitochondrial membrane, exposing both enzymatic domains to the cytosol [24].

CoA and dPCoA have multiple ionizable sites that give them a charged nature. As a result, transporters have been discovered that can exchange CoA and its synthesis intermediates between different cellular compartments. In particular, SLC25A42 can transport CoA and dPCoA from the cytosol to the mitochondrial matrix [27,28]. SLC25A16 may be another mitochondrial transporter for CoA/dPCoA [29,30], whose role is still uncertain. Another transporter was also believed to be able to deliver PPanSH directly into the mitochondrial matrix [26].

Another effect of CoA compartmentalization is its different concentrations in different cellular compartments. The mitochondrion is the organelle with the highest amount of CoA, which ranges between 2.2 and 5 mM. The peroxisome is the other compartment with a high concentration of CoA, with a value of about 0.7 mM. As a consequence, the two compartments are the most involved in CoA utilization with the central role of the cofactor in the tricarboxylic acid cycle (TCA) and fatty acid β-oxidation [21].

The numerous chemical reactions can occur precisely thanks to the highly reactive thiol group: CoA uses it to make different thioester derivatives and activate carbonyl-containing compounds [1]. Its structural conformation, dimensions, and polarity have consequences for transporting the molecule between the cellular compartments and determining its functions in cellular metabolism. The involved metabolic pathways are diverse, such as the synthesis and oxidation of fatty acids, TCA cycle and energy production, lipids and amino acids metabolism, and ketogenesis (Figure 2) [18].

Moreover, CoA and its derivatives are also involved in protein regulation by post-translational modifications (PTM) [31] (Figure 2).

The most common CoA derivative is acetyl-CoA, which is a metabolic intermediate in the production of energy from the catabolism of carbohydrates, lipids, protein, and ethanol. The acetyl-CoA/CoA ratio reflects and controls the general energetic state of the cell. However, acetyl-CoA acts as the most common intracellular acetyl group donor [9]. Numerous structural proteins and metabolic enzymes can be regulated by the transfer of the moiety to lysine residues [32]. Protein acetylation can affect its function by changing the catalytic activity, the localization, the interaction with other molecules, and its half-life [33]. For instance, the acetylation of histones, the first identified and most studied protein acetylation process, is a fundamental mechanism that controls chromatin accessibility and gene expression, utilizing the acetyl-CoA pool of the nucleus [34]. The first non-histonic acetylation substrate discovered was tubulin, a protein with a structural role in the cytoplasm [35].

CoA also mediates the covalent transfer of other acyl moieties to proteins for the dynamic modulation of their activity. Several acyl-CoA metabolites, including succinyl-CoA, propionyl-CoA, crotonyl-CoA, and palmitoyl-CoA, can be covalently attached to the side chains of lysines of proteins in different cellular compartments [36]. As for acetylation, the acylation of proteins has a role in regulating cellular processes, such as gene expression and chromatin accessibility, metabolic regulation, subcellular targeting, protein–membrane interactions, protein stability, and folding [37].

Another CoA-dependent post-translational modification of proteins is 4-phosphopantetheinylation (4PPTylation). It involves the covalent attachment of the 4′-phosphopantetheine moiety from CoA to specific serine residues in particular proteins, resulting in their activation [38]. Interestingly, the reactions in which CoA is required as an acyl transfer element do not consume it and leave it accessible for additional transfer reactions. In contrast, the addition of the 4′-phosphopantetheine group consumes CoA and releases adenosine 3′–5′-biphosphate, reducing the levels of total CoA [38]. To date, a small number of proteins in humans have been demonstrated to be activated by 4PPTylation: among them, 10-formyltetrahydrofolate dehydrogenase (FDH), involved in folate metabolism [39], cytosolic fatty acid synthase (FAS), and the mitochondrial acyl carrier protein (mtACP) [38]. Recently, 4PPTylation of mtACP has received attention as a result of the evidence linking it to the development of NBIA. Activation of the apo-mtACP protein through the 4PPTylation leads to holo-mtACP that acts in mitochondria in the assembling of respiratory chain complexes, lipoic acid metabolism, RNA processing, mitoribosome activity, and iron–sulfur cluster production [40]. The reduced activation was proposed as a possible cause of neurodegeneration in NBIA [41].

Recently, a novel redox regulation-based protein modification mechanism named CoAlation has been described, which involves the covalent modification of cellular proteins by CoA. In oxidative and metabolic stress conditions, CoA may function as a low molecular weight antioxidant preventing the overoxidation of protein cysteine thiols [42]. Additionally, CoAlation modifies the molecular mass and charge of proteins, which may affect their stability, subcellular localization, and enzymatic activity [43].

## 3. Inborn Errors of CoA Biosynthesis

Given the central role of CoA in cellular metabolism and signaling, human diseases linked to alterations in its biosynthetic pathway are very rare. To date, variants in enzymes catalyzing all the five CoA biosynthetic steps have been associated with recessively inherited conditions (Table 1).

### 3.1. PANK2

*PANK2* was the first gene associated with a genetic disorder of CoA biosynthesis [2]. Autosomal recessive pathogenic variants in this gene, which maps in humans on chromosome 20p13 and codes for a protein of 570 amino acids, cause pantothenate kinase-associated neurodegeneration (PKAN, OMIM # 234200), such as the most common NBIA form. With an estimated prevalence of 1–2/1,000,000 [44], this neurodegenerative disease is historically divided into classic and atypical forms, characterized by early childhood or late adolescence-early adulthood onset, respectively [45].

Both types are characterized by iron accumulation in *globus pallidus* (GP), which appears as a bilateral pallidal hypointensity with a central hyperintense signal in the T2-weighted MRI images of the brain. This peculiar signal takes the name “eye of the tiger sign”, and it is considered the pathognomonic sign of the disease [46], even though it can either be absent [47] or present in other NBIA [48].

Classic PKAN is characterized by early onset dystonia, spasticity, choreoathetosis, pigmentary retinopathy and neuropsychiatric symptoms, and/or dementia. These symptoms and signs are progressive and lead to a loss of ambulation (usually within 10–15 years of onset) and global motor and cognitive regression [49].

An atypical presentation, with onset in a second and third decade, is characterized by a more gradual progression of the disease. Speech defects, spasticity, and psychiatric disturbances also dominate in the atypical form; ambulatory loss occurs within 15–40 years of disease onset [49].

The neurohistological investigation of postmortem PKAN patients showed signs of pathology predominantly in the GP, although some cases displayed an involvement of other brain regions [50,51]. Main histopathological findings include profound rarefaction and vacuolization due to neuronal loss, neuroaxonal dystrophy and axonal swelling, the presence of large spheroidal structures, and an accumulation of an abnormally ubiquitinated protein. Signs of neuroinflammation, reactive astrogliosis, and activated microglia were found in GP, caudate/putamen, and neocortex. Perl’s staining demonstrated the cytoplasmic accumulation of iron in degenerating neurons and reactive astrocytes in GP [50,51].

PANK2 protein, which localizes into the mitochondrial intermembrane space, is composed of three functional regions, such as the N-terminal mitochondrial targeting sequence (MTS), the central intermediate/regulatory domain, and the C-terminal catalytic core [52,53,54].

Variants in *PANK2* gene are predicted to cause faulty CoA biosynthesis, which may result in several metabolic abnormalities, including defects in energy production, an increase in oxidative stress, and an alteration of phospholipids biosynthesis and cellular membranes remodeling [54,55,56]. This is supported by the metabolic profiles of PKAN plasma samples showing increased levels of mitochondrial dysfunction markers and decreased levels of different lipids [57]. However, CoA deficiency in human patients has not yet been proven [58].

Among the earliest tools used to study the pathophysiology of the disease were primary fibroblasts derived from PKAN patients. They displayed an altered response to iron addition, with a decreased iron internalization in ferritin and an elevated labile iron pool, which is supposed to be the cause of the elevated oxidative stress [59,60,61]. Patient fibroblasts have also shown defects in mitochondria, such as decreased membrane potential, enlarged mitochondria with altered cristae, and damaged membranes [60].

The knockdown of *PANK2* in different mammalian cells has shown an alteration in the levels of proteins involved in intracellular iron homeostasis, such as the induction of ferroportin, the only known cellular exporter of iron, as well as a decrease in ferritin (Ft) and an increase in transferrin receptor 1 (TfR1) levels [62]. More recently, glutamatergic neurons obtained from patients-derived human induced pluripotent stem cells (hiPSC), revealed abnormal iron metabolism, impaired mitochondrial respiration, altered electrophysiological activity, oxidative stress, and augmented mortality, but without any sign of evident iron accumulation [63]. Furthermore, in the same cellular model, calcium phosphate aggregates and disturbed calcium homeostasis were discovered in the mitochondria [64], characteristics observed also in vivo in some PKAN patients [64,65]. However, PKAN hiPSC-derived GABAergic neurons and astrocytes showed the presence of electron-dense granules of iron, as well as altered mitochondria, oxidative stress, and signs of ferroptosis [66].

The yeast model proved to be a useful tool for learning more about the pathophysiology of PKAN. In *Saccharomyces cerevisiae*, the deletion of the unique *pank* gene, as well as its replacement with a *PANK2* human-mutant form displayed iron overload, altered lipid metabolism, mitochondrial dysfunction, and oxidative damage [67,68].

Numerous invertebrate and vertebrate animal models have been generated and characterized to better unravel the pathogenic processes underlying PKAN and to test potential therapeutic options.

Due to the existence of a unique PANK gene (*fumble*) in *Drosophila melanogaster*, it is possible to study the effects of its lack without a compensatory mechanism derived from other isoforms. *Drosophila*-derived S2 cells silenced for the *pank/fbl* gene exhibited several of the characteristics described in mammalian cells. In addition to decreased growth, mitochondrial dysfunction, and greater vulnerability to reactive oxygen species (ROS), these cells also displayed a considerable drop in CoA content, disturbed F-actin filament organization, less effective DNA repair pathways, and a reduction in the acetylation of histones and tubulin [23,69,70].

The fly *pank*/*fbl* mutant model exhibited locomotor dysfunction, altered wing morphology, infertility, and a reduced lifespan, as well as an increased apoptosis in the larval brains, abnormal lipid metabolism with disruption in lipid biosynthesis, and neurodegeneration. Moreover, flies presented a reduction in CoA content, movement alterations, and increased sensibility to oxidative stress that led to an increased DNA damage and a reduced life span [69,71,72]. Prominent mitochondrial aggregations and reduced ATP levels were found in dopaminergic neurons of *fbl* knockdown flies [73]. Interestingly, a functional interaction between *fbl* and the mitochondrial quality control regulator PTEN-induced serine/threonine kinase 1 (*PINK1*), dependent on intracellular CoA and acetyl-CoA levels, has been proposed [73]. In fact, it has been suggested that *fbl* mRNA follows a PINK1/Parkin-regulated translational mechanism that is localized on the mitochondrial membrane. Moreover, Fbl availability regulates p62/SQSTM1 homolog by acetylation to promote mitophagy, thus affecting mitochondrial quality control [73].

The generated animal models also involve *Danio rerio*, which possesses one ortholog of *PANK2* and three other paralogs (*pank1a*, *pank1b*, and *pank4*). To better understand the role of the *pank2* gene in zebrafish development, a model was created by injecting a splice-inhibiting morpholino. The obtained morphants showed a phenotype characterized by an altered development of central nervous system (CNS), as well as an activation of neuroinflammation and neuronal loss in the telencephalon, corresponding to human GP [74]. Another recent study in zebrafish identified a link between pank2 abnormalities and the process of angiogenesis [75]. Moreover, *pank2* downregulation with a concomitant overexpression of human PANK2 mutant forms in zebrafish embryos led to an alteration of the vascular structures, especially in the caudal plexus and in the tail [76]. Recently, a genetic model of *pank2* knockout (KO) in zebrafish has revealed abnormalities in the development of venous vascular structures in mutant embryos and testicular atrophy in adult fish [77].

The first generated mammalian model of the PKAN disease was the *Pank2* KO mouse [78]. The phenotypic manifestation of the animal does not include the neurological symptoms observed in humans. MRI and Perl’s staining showed no signs of iron accumulation, and no differences in tests for the assessment of neurological impairment were observed between wild-type (WT) and KO animals. However, the mice exhibited complete male infertility due to a block in spermatogenesis, as well as a reduced body weight and moderate retinopathy in very old KO animals. Ex vivo studies highlighted a global alteration of mitochondria in the CNS, characterized by impaired respiration, swollen shape, and aberrant cristae [79]. Various attempts have been made to exacerbate a possible neurological phenotype. Interestingly, feeding the mutant mice with a Pan-deficient diet led to a reduced lifespan and the appearance of mild neurological anomalies [80]. Furthermore, KO mice treated with a ketogenic diet developed a neuromuscular phenotype, characterized by motor dysfunction, neurodegenerative signs, and a structural alteration of muscle morphology [81].

Recently, an in-depth re-evaluation of this *Pank2* KO model was performed by focusing the investigation on the GP-containing region. This analysis highlighted the perturbation of iron homeostasis, mitochondrial functionality, and dopamine metabolism. In addition, the downregulation of mRNA of two enzymes downstream to Pank2, i.e., Coasy and Ppcs, was observed in both the brain and blood, and was suggested to be a possible biomarker of disease. However, the mice displayed neither neurological symptoms nor neurodegenerative signs [82]. The reason for the lack of a human-related phenotype in the *Pank2* KO mouse model is still missing. However, whereas PANK2 mitochondrial localization in humans is generally accepted, in mice, this notion is still widely debated [24,79]. In addition, it has been suggested that the murine PANK isoforms are able to compensate each other to maintain homeostatic levels of CoA. Indeed, the activity of the various isoforms appears to be redundant with this evolutionary purpose. Different murine models were generated to better understand the roles of the various Pank isoforms. Reduced liver CoA and altered fatty acid oxidation were the main metabolic abnormalities found in the *Pank1* KO animal [83]. The constitutive double *Pank1*/*Pank2* KO mouse showed acute hypoglycemia and hyperketonemia that resulted in a severe impairment of postnatal development and early death by postnatal day P17. It also showed hepatic symptoms with lipid accumulation and compromised fatty acid oxidation [84]. Furthermore, the conditional neuronal-specific *Pank1*/*Pank2* double KO model exhibited a median survival of 52 days and a significant reduction in CoA brain content [85], while a mouse model with a systemic deletion of *Pank1*, together with the neuronal deletion of *Pank2*, had a median lifespan of 17 days and a significant drop of brain CoA levels [86].

The precise molecular mechanisms linking CoA biosynthesis alteration with neurodegeneration and brain iron accumulation have not yet been established. However, some possible mechanisms underlining the pathogenesis of the disease have recently been suggested. As mentioned before, CoA levels have been correlated with the activation of the mtACP by 4PPTylation, and this reaction consumes the intracellular pool of CoA. Lambrechts et al. [41] have demonstrated in mammalian cells and *Drosophila* that the reduction in CoA levels can affect the 4PPTylation of mtACP, which has a role in mitochondrial fatty acid synthesis and iron–sulfur cluster formation. A decrease in CoA levels causes the defective 4PPTylation of mtACP, leading to a defect in the synthesis of lipoic acid in mitochondria. Pyruvate dehydrogenase (PDH) complex needs lipoylation in its dihydrolipoamide acetyltransferase (PDH-E2) subunit to function properly, and the lack of this activation leads to a decrease in activity. As a consequence, mitochondrial health is affected, considering the role of PDH in catalyzing the transformation of glycolysis-derived pyruvate into acetyl-CoA, the essential step for the beginning of the TCA cycle. Further indirect evidence correlating the PANK2 deficiency with the mtACP low functionality was the reduction in PDH and complex I activities observed in the GP of the *Pank2* KO mouse and fibroblasts derived from PKAN patients [82,87]. In addition to PDH, other mitochondrial enzymes are modified by lipoylation, among which are α-ketoglutarate dehydrogenase (αKGDH) and branched-chain α-keto acid dehydrogenase (BCKDH) [88]. These enzymes are presumably affected as well, as indicated by decreased levels of lipoylated protein levels in yeast [89], Drosophila [41], and human [87] PANK2-deficient cells, although their enzymatic activities have never been measured yet.

Another proposed mechanism involved in the PKAN disease is the complex balance in iron intracellular homeostasis and trafficking. It has been observed that when exposed to high iron levels in the culture medium, fibroblasts from PKAN and other NBIA patients readily accumulate more iron compared to control cells. This phenomenon was demonstrated to be a consequence of impaired internalization and the recycling of the TfR1 receptor that remains on the cellular membrane and continues to promote iron uptake. The occurrence was attributed to improper protein palmitoylation, which altered protein trafficking and recycling [90]. Moreover, PKAN astrocytes have also shown a compromised endosomal compartment functioning with an increase in cellular transferrin uptake. It was proposed that a CoA-dependent impairment of vesicular trafficking would cause unfavorable iron accumulation in astrocytes, which might lead to a series of events that would result in neuronal death in PKAN patients [91]. These data suggest a common pathogenic pathway shared by NBIA disorders with different genetic causes.

However, despite the considerable efforts in the generation of animal and cellular models, there are still considerable gaps in our understanding of the etiology of this pathology.

### 3.2. COASY

COASY is the bifunctional CoA synthase catalyzing the two last reactions of the CoA biosynthesis. In humans, the *COASY* gene is located on chromosome 17q21.2 and encodes a protein of 564 amino acids [92]. The existence of an alternative isoform has been shown, named COASY β, generated by alternative splicing and possessing an additional 29 amino acids at its N-terminus compared to the canonical COASY α [93]. Moreover, in contrast to COASY α, which is ubiquitously expressed, COASY β has been reported to be expressed only in the kidney, heart, and brain. Both COASY isoforms are localized primarily on the mitochondrial outer membrane and in the mitochondrial matrix, although a free cytoplasmic and nucleic localization has been reported [24,25,93,94].

Recessive pathogenic variants in *COASY* were firstly linked to a disorder whose clinical symptoms are partially overlapping with those of PKAN [3]. This condition was denominated as COASY protein-associated neurodegeneration (CoPAN, OMIM # 615643) and is very rare, with an estimated prevalence of less than one in a million [95].

The clinical picture of CoPAN is characterized usually by a normal early psychomotor development followed by delayed or abnormal walking due to the spasticity of the lower limb. The disease is progressive, with a worsening of the motor signs in the lower limbs and a progressive involvement of the upper limbs and oro-mandibular region that can be affected by dystonia. The severe clinical picture is characterized by spastic-dystonic tetraparesis with a prevalent involvement of the lower limbs and parkinsonian features. Axonal motor neuropathy causing distal amyotrophia and areflexia with pes cavus can be present. Psychiatric disturbances such as obsessive-compulsive disorder and/or depression can occur [3,96,97].

Thereafter, *COASY*-recessive loss-of-function variants have been found in thirteen fetuses/neonates from seven unrelated families with pontocerebellar hypoplasia type 12 (PCH 12). These variations cause a severe prenatal onset phenotype characterized by an atrophy of the pons and cerebellum, microcephaly, and arthrogryposis, which is invariably fatal in the perinatal period. However, no signs of iron accumulation in the brain were observed in these affected individuals [98,99].

Recently, an intermediate clinical condition was described in two patients who had hypotonia and respiratory failure. Basal ganglia and brainstem atrophy were also seen on magnetic resonance imaging, along with gradual widespread parenchymal loss in both cerebral hemispheres, while no signs suggestive of iron accumulation were observed. Electromyography revealed substantial muscular degeneration in the form of abundant fibrillation potentials in the tested muscles [100].

Therefore, the expansion of the clinical features opens the possibility of overlooked symptoms associated with variants in this gene. Interestingly, in females with Down syndrome, a *COASY* exon 4 single-nucleotide polymorphism (SNP) has been proposed to be a risk factor for early onset Alzheimer’s disease (AD) [101].

Although only a few cases mutated in *COASY* have been published, a recent study analyzing the lifetime risk of 13 autosomal recessive NBIA disorders calculated from the sum of allele frequencies reported in genetic databases reveals that CoPAN is among the most frequent NBIA forms, ranking third after PLAN (PLA2G6-associated neurodegeneration) and PKAN, with a frequency of pathogenic *COASY* alleles of about 0.0012. Importantly, this approach represents lifetime risks from conception, thus including prenatal deaths [102].

So far, few COASY-associated models have been generated and characterized. In *Saccharomyces cerevisiae*, the two PPAT and DPCK activities are catalyzed by two different enzymes encoded by *CAB4* and *CAB5* genes, respectively. The deletion of both genes resulted in a lethal phenotype, which was reverted by the re-expression of the human COASY protein. Moreover, the expression of a human-mutated *COASY* showed a reduction in mitochondrial CoA, mitochondrial dysfunctions with a reduction in mitochondrial respiration, more sensitivity to the ROS-inducing agent, and iron accumulation [103,104].

In *Zebrafish*, it was possible to achieve different *coasy* gene residual expression levels by varying the morpholino dosages. A dorsalized mutant-like phenotype resulted from the total abrogation of its expression, which also caused a significant decrease in CoA content, a lack of normal embryonic development, and high mortality. Instead, lower morpholino dosage resulted in a reduction in *coasy* expression of about 60%, as well as the onset of a milder phenotype. Most of the embryos showed almost normal development at 24 h post-fertilization (hpf) but were characterized by a poorly defined brain structure and head edema at 48 hpf [105].

Since constitutive ablation of *Coasy* is incompatible with life, the first, and to date, the only mammalian model generated was a conditional neuronal-specific *Coasy* mouse KO model [106]. The animals showed a severe early onset neurological phenotype starting with a growth arrest from the age of postnatal day 8, locomotor abnormalities, and dystonia-like movements. Autoptic analysis of the brain showed mitochondrial structure and function alterations, as well as iron dyshomeostasis characterized by a significant increase in the cytoplasmic ferritin light chain (FtL) and a reduction in both transferrin receptor 1 (TfR1) and divalent metal transporter 1 (DMT1). However, although the Coasy protein was reduced by 50%, CoA levels in total brain homogenates were comparable to those of control littermates [106]. Interestingly, a similar scenario was seen also in primary fibroblasts derived from CoPAN patients, which showed a reduction in the rate of CoA biosynthesis, while the amount of total CoA was comparable to controls [3]. This evidence has raised some questions about the existence of alternative enzymes or biochemical pathways for CoA biosynthesis.

Studies on the functions of COASY have also been carried out in the oncological field for the role of CoA and acetyl-CoA in the regulation of cellular proliferation, or to identify potential biomarkers for early diagnosis. Recently, *COASY* was proposed as a new biomarker as its mRNA expression was consistently enhanced in radioresistant human rectal tumors. The mechanism involves the activation of the PI3K signaling pathway and the increase in AKT and mTOR phosphorylation, enhancing cell survival [107]. Moreover, a physical interaction between the ribosomal protein S6 kinase (S6K1), which is under the control of PI3K and mTOR signaling pathways, and COASY was reported [108]. However, in triple-negative breast cancer cell lines, the knockdown of *COASY* resulted in no impact on cell proliferation [109]. *COASY* was seen to also have a role in mitotic regulation: some evidence indicates that acetylation ensures faithful mitosis, and the *COASY* knockdown triggers prolonged cellular division [94].

The investigations of potential COASY interactors have led to interesting results that could help understanding the pathogenesis of the diseases caused by variants in this gene. COASY has been found to form a functional complex with p85αPI3K, and changes in PI3K signaling pathway activity have been seen to occur after COASY silencing [110]. Moreover, COASY was found to interact with many signaling proteins, including the Shp2 tyrosine phosphatase, which regulates COASY PPAT catalytic activity, or with the enhancer of mRNA-decapping protein 4 (EDC4), which inhibits the DPCK activity and, therefore, CoA production and cell proliferation [111,112].

Recently, some studies have correlated the *COASY* gene promoter methylation with the pathophysiology of Alzheimer’s disease (AD) and amnestic mild cognitive impairment (aMCI), indicating that this parameter could be a useful blood prodromic biomarker [113,114].

### 3.3. PPCS

PPCS, the second enzyme of the CoA biosynthetic pathway, is encoded by a gene that in humans is located on chromosome 1p34.2 and codes for two isoforms: the canonical isoform of 311 amino acids, and a shorter isoform of 138 residues. Both of them are ubiquitously expressed and have the C-terminal region in common [21].

Recently, pathogenic variants in the *PPCS* gene were associated with a human disease through the identification of five individuals from two unrelated families [6]. Exome sequencing revealed heterozygous or homozygous biallelic sequence variants in the *PPCS* gene, suggesting a recessive inheritance. The affected subjects were characterized by infantile onset dilated cardiomyopathy of a variable degree of severity, in one case associated with cutis laxa and dysmorphic features, without evident signs of neurological involvement. Moreover, the *PPCS* variants led to a significant reduction in CoA content in patients’ fibroblasts compared with healthy controls, which was restored by the transduction of the wild-type *PPCS* gene [6]. Another case has been reported about a female infant, who was compound heterozygous with a missense variant and an intronic splice site variant [7]. The measure of the CoA content in patient fibroblasts was significantly reduced. In addition to dilated cardiomyopathy, the clinical symptoms also included hypotonia at birth, dysmorphic features, necrotizing myopathy and enterocolitis, and intermittent rhabdomyolysis. Her initial metabolic screening for urinary organic acids, plasma amino acids, very long-chain fatty acids, and a blood spot acylcarnitine profiles were reported as normal [7]. Iuso and collaborators generated a *Drosophila melanogaster* knock-down model, which showed significant alterations in the cardiac parameters, such as an increase in heart rate and a decrease in systolic length with neurological involvement [6]. Subsequently, the same group reported the generation of two hiPSC cells carrying pathogenic biallelic variants in the PPCS gene [115].

To date, no mammalian animal model of PPCS-associated dilated cardiomyopathy has been described.

### 3.4. PPCDC

PPCDC is the third enzyme of the CoA biosynthetic pathway. The gene is located on chromosome 15q24.2, encoding a protein of 204 amino acids [116]. Recent research has linked variants in the *PPCDC* gene to human pathology. Two sisters reported to share biallelic missense variants showed an early phenotype with clinical manifestations starting a few days after birth. They presented fatal dilated cardiomyopathy, as well as lactic acidosis, elevated creatine kinase, hyperammoniemia, liver disease, and a neurological involvement with lethargy and limbs’ hypertonia. The echocardiographic evaluation showed a systolic dysfunction and a decrease in ejection fraction that result quickly in fatal outcomes [8]. Biochemical analysis in body fluids revealed hypoglycemia, ketonuria, alanine elevation, urinary excretion of dicarboxylic acids, and increased levels of long-chain acylcarnitine in plasma samples, all suggestive of a defect in fatty acid oxidation (FAO) or mitochondrial disease [8].

Functional analysis and molecular investigations were conducted in patients-derived fibroblasts and a yeast model of *Saccharomyces cerevisiae.* Patient-derived fibroblasts showed a complete absence of the PPCDC protein, intracellular CoA levels of about 50% of healthy controls, and defects in mitochondrial respiration. Moreover, the evaluations conducted in yeast confirmed the pathogenicity of the variants [8].

## 4. Proposed Therapeutic Options

Although the first patients with alterations in the CoA biosynthetic pathway were described more than 20 years ago, no disease-modifying therapeutic option has yet been approved. Numerous attempts have been made (Figure 3), but the chances of success are low due to the still unclear pathogenic mechanisms. As for PKAN, the most common amongst the NBIA group, as well as for the other forms, the current treatments are based on the management of the symptoms [117].

Most of the experimental options discussed in the following paragraphs have shown evidence of efficacy in PKAN models or patients [118]; however, little is known about the other CoA biosynthesis-related pathologies because of the limited availability of models and the rarity of the diseases.

### 4.1. Metabolic Supplementation

Although many of the studies conducted in cells and animals have not found defects in CoA levels, many potential therapeutic approaches are based on attempts to increase intracellular CoA content. Various CoA precursors or intermediates have been used to achieve this goal.

#### 4.1.1. Pantothenate

The use of pantothenic acid was based on its potential capacity to boost the PANK2 enzymatic residual activity by increasing substrate availability. This option was proposed for the treatment of atypical patients who are supposed to have pathogenic variants that mostly preserve enzymatic activity. Preliminary evidence of efficacy was observed in fibroblasts derived from patients and in induced neurons with residual PANK2 activity [61]. It has been demonstrated that pantothenate can correct the labile iron pool levels and iron accumulation. No effect was observed when the protein expression was completely abolished [61]. Moreover, pantothenate can correct the defect in the expression of mitochondrial phosphopantetheinyl-proteins such as mtACP [87] and reverse some pathological alterations observed in PKAN fibroblasts [119]. Similar results were obtained in Zebrafish [74]. However, many clinicians reported no effects in the use of pantothenate as a nutritional supplement in PKAN patients (personal communications), although an ad hoc clinical trial is missing. It has been proposed to detect pantothenate-responder patients through an analysis of their cells before the beginning of treatment with pantothenate [120].

#### 4.1.2. Pantethine

Pantethine is the stable disulfide form of PanSH. After intake, the molecule is immediately hydrolyzed to PanSH, which can be phosphorylated to form PPanSH, which is the substrate for the COASY enzyme [121]. In this way, the molecule can bypass the defective steps upstream of the COASY enzyme. Evidence of its efficacy was provided in different *PANK2* models starting from patients’ fibroblasts. Here, a partial rescue of phenotype was observed after pantethine treatment [119]. In the *Drosophila* model, pantethine can recover almost completely the phenotype seen in the hypomorph mutant *dPANK*/*fbl* [23]. Moreover, the defects in histones and tubulin acetylation levels observed in the *Drosophila* model were restored after pantethine administration [70]. In zebrafish embryos, the altered phenotype appears to be restored by exposing the *pank2* morphant embryos to pantethine [74]. Additionally, pantethine feeding is efficient in reducing the pathological phenotype seen in *Pank2* KO mice fed with the ketogenic diet [81].

Pantethine has been also used on the fly model of PPCS as a food additive [6]. Subsequently, three of the six PPCS patients reported having received oral pantetheine supplementation, but with contrasting and unclear results due to the severity of the phenotype and the treatment’s late introduction [6,7]. The use of pantethine in one PPCDC patient has provided encouraging indications [8]. The effectiveness of pantethine was observed also in a model of *Escherichia coli* in which the gene *coaBC*, which encodes a bifunctional protein that has PPCS and PPCDC function, was deleted [122].

A single-arm, open-label study was conducted in children with PKAN to assess the efficacy and safety of pantethine (ChiCTR1900021076). After 24 weeks of treatment, the results showed that pantethine is well tolerated but it did not significantly improve motor function, although in some cases it can slow down the progression of motor dysfunction [123].

#### 4.1.3. Fosmetpantotenate

Fosmetpantotenate (RE-024) is a PPan prodrug designed to restore PPan levels in PKAN patients. The membrane permeability is elevated thanks to a charge-masking on the phosphate dianion, and once in the cell, it can become a substrate for the downstream enzyme of the pathway [124]. In cells with silenced *PANK2*, fosmetpantotenate restored CoA levels and rescued the defects in tubulin acetylation. In vivo studies have proven good pharmacokinetics and tissue delivery [124]. The first patient treated with fosmetpantotenate, characterized by a late-onset moderately severe phenotype, showed a clinical improvement in all the analyzed parameters after 12 months of treatment [125]. However, a successive randomized controlled trial on 84 patients demonstrated its safety, but it did not improve the patients’ clinical condition [126,127]. Moreover, cyclic phosphopantothenic acid prodrugs have been developed in order to obtain more stable and more efficacious compounds [128].

#### 4.1.4. Phosphopantetheine and Acetyl-Phosphopantetheine

PPanSH has been proposed as a therapeutic option for PKAN condition. It has been demonstrated that PPanSH is stable in serum and in the circulation of animals for several hours, and is able to cross membranes in a transporter-independent way [22]. In *C. elegans*, *D. melanogaster*, or cultured mammalian cells, intracellular CoA levels are restored via PPanSH uptake, followed by its conversion into CoA by the two final enzymes of the canonic CoA biosynthetic pathway [22].

Recently, the group of Susan Hayflick demonstrated that PPanSH, as a diet supplement, can reverse some molecular anomalies found in the GP of *Pank2* KO mice [82]. Those data led to the initiation of a clinical trial (NCT04182763), which is currently ongoing.

Acetyl-4′-phosphopantetheine (PPanSAc), the acetylated form of PPanSH, has shown results in preventing or reversing the PKAN-related phenotype in flies and mice [129].

These two strategies can be used when the blockage of CoA biosynthesis is due to variants in *PANK2*, *PPCS,* or *PPCDC*, but not when the mutated gene is *COASY*.

#### 4.1.5. Coenzyme A

CoA supplementation was seen to partially or completely rescue the pathological phenotype observed in PKAN neurons or astrocytes differentiated from patients-derived hiPSC [63,66,91], as well as to correct TfR1 palmitoylation in fibroblasts derived from PKAN patients [90]. It was also effective in recovering cell counts in *dPANK*/*Fbl*-depleted S2 cells [23]. Moreover, exogenous CoA efficacy was demonstrated in other PKAN cellular and multicellular models, such as worm, fly, and zebrafish [22,74]. However, it was postulated that the CoA is unable to pass membranes and that it is rapidly degraded into PPanSH by NUDT and ENPPs enzymes. PPanSH could pass the cellular membrane and become the substrate of the COASY enzyme [130]. In contrast, some results have demonstrated that the addition of CoA to fish water can restore the normal development in the *coasy* model of zebrafish [105].

### 4.2. Allosteric Activators of PANKs Isoforms

The investigations on PANK activators have opened the way to an innovative approach based on the stimulation of the other PANK enzymes when PANK2 is defective. Among the many compounds found [131], those with characteristics of polarity, lipophilicity, and molecular weight suitable for crossing the blood–brain barrier (BBB) were selected. The candidate molecule was the pantazine PZ-2891, which is able to reach the CNS, cross the cellular membrane, and function as an allosteric activator avoiding the feedback inhibition of the PANK enzymes by CoA or acetyl-CoA. It was effective in rescuing the severe locomotor phenotype in a mouse model in which both *Pank1* and *Pank2* genes were deleted. Moreover, it was able to increase brain CoA content after oral administration [85]. A first-in-human trial to assess the safety and tolerability of an improved pantazine (BP-671) in healthy volunteers has recently begun (NCT04836494). The molecule was also proposed as a therapeutic option for patients with Propionic Acidemia and Methylmalonic Acidemia [132].

### 4.3. Improvement of Mitochondrial Function

Some cell and animal models of PKAN show a phenotype characterized by mitochondrial dysfunction [66,81]. A recent study has validated the efficacy of Leriglitazone, an agonist of PPARγ (Peroxisome proliferator-activated receptor gamma), in correcting the pathological phenotype in PKAN astrocytes [133]. PPARγ is a crucial regulator of inflammation, energy metabolism, and anti-oxidant defense, and supports mitochondrial function and biogenesis [134]. Leriglitazone is a derivate of pioglitazone able to pass the BBB and its efficacy was observed also in X-linked adrenoleukodystrophy (X-ALD) and Friedreich’s ataxia (FRDA) [135,136]. In PKAN astrocytes, Leriglitazone was able to ameliorate cell viability, improve mitochondrial functions, and reduce iron accumulation [133]. The safety and efficacy in patients with Friedreich Ataxia or X-linked adrenoleukodystrophy were already evaluated in two different clinical trials [137,138].

### 4.4. Iron Chelators

The pathognomonic characteristic of PKAN and CoPAN was the iron accumulation in the brain. Its deleterious effect on cellular homeostasis comes from the ability to induce oxidative stress by Fenton’s reaction, which produces ROS. The oxidized adducts produced by their reaction with proteins, lipids, and DNA have a damaging effect that can result in cell death. In particular, the accumulation of peroxidized lipids can induce the process of cell death named ferroptosis [139].

How the process of accumulation takes place in the brain of patients is still unclear, and the available models have not been able to clarify this aspect because of their distance from the human disease context. The suggested hypothesis is based on the defective post-translational modifications of the proteins involved in iron metabolism [90].

The proposed use of iron chelators was prompted by the attempt to reduce the deleterious accumulation in patients’ brains. The molecular functioning of the iron chelators is based on the binding to form a stable complex that can be easily excreted from the body. The FDA-approved molecule deferiprone (DFP) has shown the ability to cross the BBB, and for this reason, its efficacy was tested in patients with NBIA [140]. It was first used to treat iron excess caused by blood transfusion administration and to treat hematological pathologies, including thalassemia. The medication was well tolerated but recent evidence has demonstrated that, even though patients’ iron deposits did significantly decrease after taking the medication, the symptomatology did not have a significative improvement, although a slowdown in the disease progression was observed [141,142]. Given the lack of other therapies and the fact that iron causes at least a part of the pathophysiology, the documented DFP tolerability supports iron chelation as a therapy for PKAN [143] patients.

## 5. Conclusions

Although CoA metabolism has been known for a long time, the discovery of new pathologies associated with its synthesis has brought it back into the spotlight. Defects in CoA biosynthetic pathway have been associated with four genetic diseases in humans. Two of them, PKAN and CoPAN, are neurological disorders belonging to the group of neurodegenerations with brain iron accumulation (NBIA) characterized by iron overload in specific brain areas. The other two are dilated cardiomyopathies characterized by early onset and premature death. The two clinical manifestations are very different, and a possible explanation could be sought by studying the common features of the two groups of genes. For example, the subcellular localization of the involved protein: PANK2 and COASY are strongly connected with mitochondria, while PPCS and PPCDC have cytoplasmic localization. The investigation of this point could help to clarify some features of the pathophysiology of the four diseases and consolidate the very preliminary data on PPCS- and PPCDC-related cardiomyopathies.

The many models generated have played a key role in improving knowledge, although many features are still unclear, e.g., why the alterations of PANK and CoPAN lead to iron overload in the human brain.

Furthermore, the development of effective therapies remains a challenge and cannot be accomplished without a thorough understanding of the pathogenetic mechanisms underlying the diseases.

## Figures and Tables

**Figure 1 ijms-24-05951-f001:**
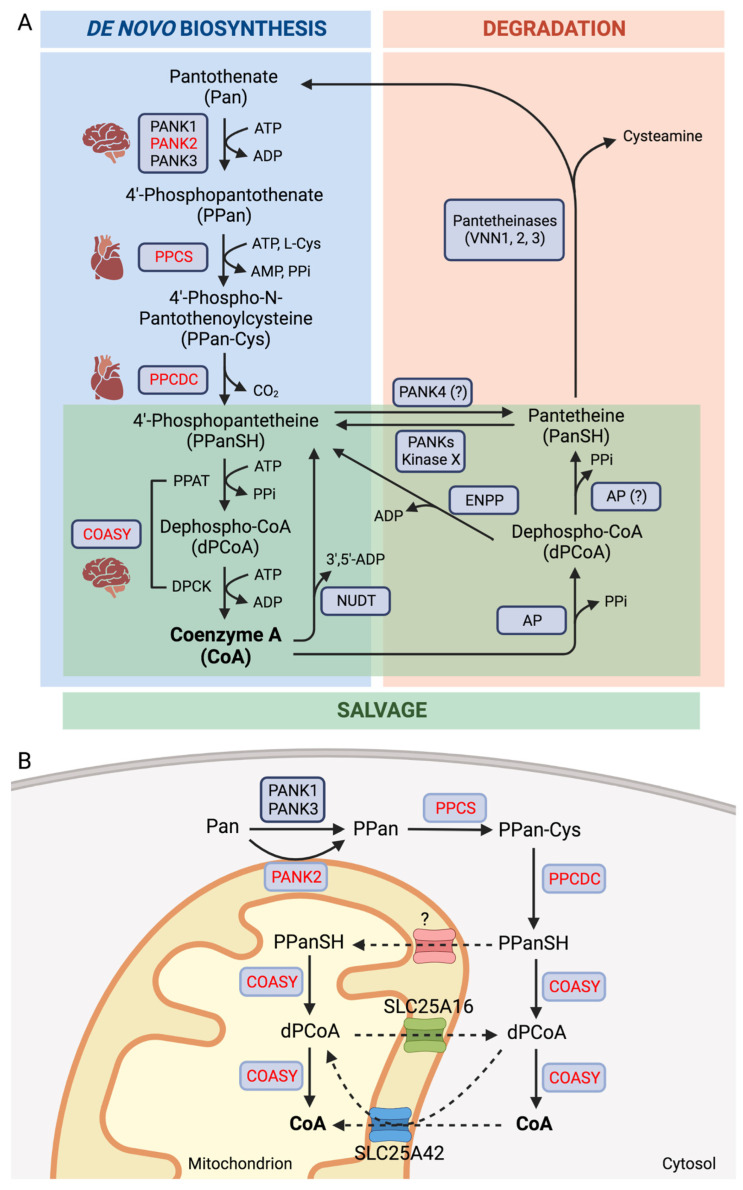
Coenzyme A homeostasis. (**A**) Schematic representation of CoA biosynthetic (blue), salvage (green), and degradation pathways. (**B**) Intracellular localization of CoA biosynthesis. Disease-associated enzymes are in red. Abbreviations: PANK, pantothenate kinase; PPCS, phosphopantothenoylcysteine synthetase; PPCDC, phosphopantothenoylcysteine decarboxylase; PPAT, phosphopantetheine adenyltransferase; DPCK, dephospho-CoA kinase; COASY, CoA synthase; NUDT, nucleoside diphosphate-linked moiety X-type motif; AP, alkaline phosphatase ENPP, ectonucleotide pyrophosphatase/phosphodiesterase.

**Figure 2 ijms-24-05951-f002:**
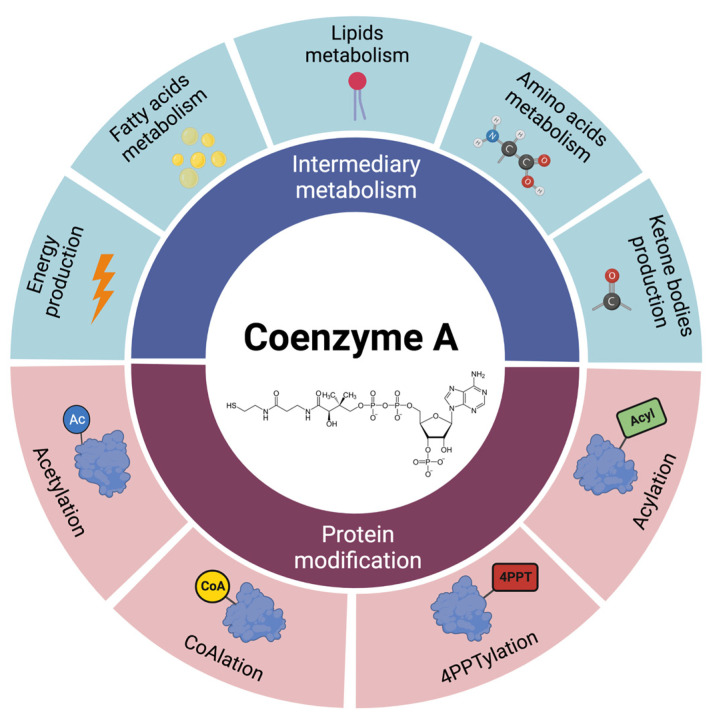
Coenzyme A cellular functions. CoA is implicated in diverse cellular activities, which can be divided into metabolic (blue) and protein modification (red) functions.

**Figure 3 ijms-24-05951-f003:**
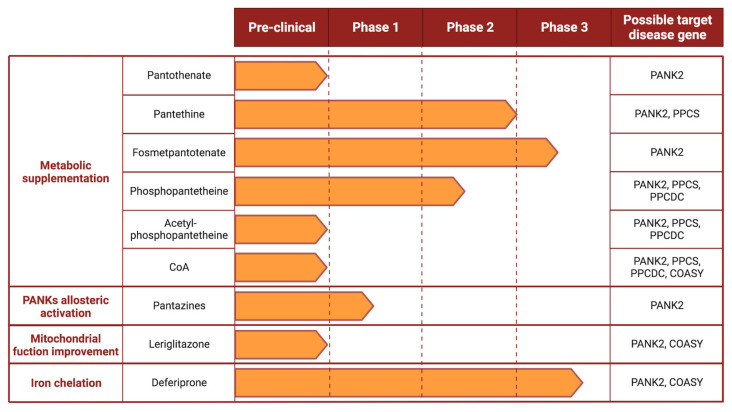
Proposed therapeutic strategies for inherited disorders of CoA biosynthesis and their current stage of development.

**Table 1 ijms-24-05951-t001:** Molecular and clinical features of inherited disorders of CoA biosynthesis.

**Gene (* OMIM)**	**Protein Function**	**Cellular** **Localization**	Disease (# OMIM)	Inheritance	Clinical Features
PANK2 (* 606157)	Pantothenate kinase	Mitochondrial intermembrane space	Pantothenate kinase-associated neurodegeneration, PKAN (# 234200)	AR	Early (classical) or late (atypical) onset dystonia, spasticity, cognitive decline, pigmentary retinopathy. Iron overload in GP (eye of the tiger sign).
PPCS (* 609853)	Phosphopantothenoyl cysteine synthetase	Cytosol	Dilated cardiomyopathy 2C, CMD2C (# 618189)	AR	Early onset dilated cardiomyopathy, hypotonia, necrotizing myopathy, intermittent rhabdomyolysis.
PPCDC (* 609854)	Phosphopantothenoyl cysteine decarboxylase	Cytosol	Dilated cardiomyopathy	AR	Early onset dilated cardiomyopathy, lactic acidosis, neurological involvement with lethargy and limb hypertonia.
COASY (* 609855)	Phosphopantetheine adenylyl-transferase (PPAT) and dephospho-CoA kinase (DPCK)	Mitochondrial outer membrane, mitochondrial matrix, cytosol, nucleus	COASY protein- associated neurodegeneration, CoPAN (# 615643)	AR	Early onset spastic-dystonic paraparesis, dysarthria, obsessive-compulsive behavior. Iron overload in GP.
			Pontocerebellar hypoplasia type 12, PCH12 (# 618266)	AR	Perinatal onset lethal microcephaly and arthrogryposis.

OMIM: Online mendelian inheritance in man, https://www.omim.org (accessed on 7 March 2023); * Gene MIM; # Phenotype MIM; AR autosomal recessive.

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
