# Peer review of "Inherited Disorders of Coenzyme A Biosynthesis: Models, Mechanisms, and Treatments"

_ijms, 2023, doi:10.3390/ijms24065951_

Round 1

Reviewer 1 Report

This is a comprehensive and well-structured review article.  The text is well written overall, and clearly understandable in most places, while the figures and table are helpful and appropriate. 
Each of my suggested corrections/revisions is minor, but overall they are intended to ensure clarity and avoid distractions for the readers of what is otherwise a nice article, with excellent scientific and clinical content. 
These comments are mostly related to English syntax/grammar, to genetic terminology, or to placement of reference citations.     

Abstract, line 12 (and again elsewhere several times in the main text) - 'encode for' should be replaced by either 'encode' or 'code for'.
line 20 - 'associated to', replace 'to' by 'with'

Introduction, line 44 - add a reference citation here for the 'discovery of affected families',
and line 46 - add reference citation here for 'condition in two sisters'.

Page 2, line 78 - replace 'differed' by 'differing'
line 94 - delete 'a',
line 97 - replace 'at the end' by 'finally'

Page 4, line 157 - replace 'peroxisomal' by 'peroxisome'
line 158 - replace 'the highest' with 'a high'

Page 5, line 167 - delete 'the most'
line 176 (paragraph about acetyl-CoA) - the authors might wish to add a citation to a recent (Jan 2023) review article about acetyl-CoA by Y. Wang et al. (Molec. Genet. Metab.) 
line 194 - after 'attachment of', insert 'the'

Page 6, line 207 - delete 'The'

Page 7, line 224 - replace 'has been' by 'was'
line 225 - 'mutations' - here, and in several more places later in the text, it would be better not to use this word, which is no longer recommended in the human and medical genetics community.  'Sequence variants', 'variants' or 'pathogenic variants' might be used instead, depending on the context.
line 255 - delete 'such as'
line 265 - replace 'One of' by 'Among' (because fibroblasts are plural)
line 271 (and again later in the text) - replace 'mammal' by 'mammalian'

Page 8, line 308 - 'dependent from' - do the authors mean 'INdependent from' OR 'dependent on' (which have opposite meanings) ?   

Page 9, line 345 - replace 'transversally' by 'generally'
line 364 - 'Lambrechts et al.' - where is the numbered reference citation for this?

Page 10, line 395 - replace 'It' by 'There'
line 397 - replace 'N-termini' by 'N-terminus' or 'N-terminal'
line 422 - delete 'The'
line 434 - 'form' should be 'forms'

Page 11, line 440 - italicize the gene names

Page 12, line 502, replace 'lead' by 'led'
line 504 - replace 'which' by 'who'
line 506 - replace 'resulted' by 'was' (same thing for Page 13, line 562)
line 535 - replace 'pathogenesis' by 'pathogenicity'

Page 13, line 544 - 'others' should be 'other'

Page 14, line 574 - 'upstream to' should be 'upstream of'

Page 15, line 631 - replace 'in' by 'into'
line 632 'In this form, it' - do the authors mean PPanSH?  If so, it would be clearer to just say 'PPanSH'  
line 640 - replace 'exceeding' by 'traversing' or 'crossing'
line 665 - 'oxidated' should be 'oxidized'
line 671 - replace 'from the human sign of disease' to 'from the human disease context' (or something similar)
line 673 - replace 'urged' by 'motivated' or 'prompted'

Page 683, line 683 - replace 'takes' by 'causes'
line 687 - delete 'the'
line 692 - 'others' should be 'other'
line 700 - 'played a key role in improving the knowledge', replace by 'have played a key role in improving knowledge'

.

Author Response

We really appreciate the effort made by the reviewer to improve the manuscript, especially for the help with the English syntax/grammar. 

All the suggested corrections/changes have been included in the text as follows:

Abstract, line 12 (and again elsewhere several times in the main text) - 'encode for' should be replaced by either 'encode' or 'code for'.
line 20 - 'associated to', replace 'to' by 'with'

All done

Introduction, line 44 - add a reference citation here for the 'discovery of affected families',
and line 46 - add reference citation here for 'condition in two sisters'.

References added

Page 2, line 78 - replace 'differed' by 'differing'
line 94 - delete 'a',
line 97 - replace 'at the end' by 'finally'

All done

Page 4, line 157 - replace 'peroxisomal' by 'peroxisome'
line 158 - replace 'the highest' with 'a high'

Replaced

Page 5, line 167 - delete 'the most'
line 176 (paragraph about acetyl-CoA) - the authors might wish to add a citation to a recent (Jan 2023) review article about acetyl-CoA by Y. Wang et al. (Molec. Genet. Metab.) 
line 194 - after 'attachment of', insert 'the'

Corrected and citation added

Page 6, line 207 - delete 'The'

Deleted

Page 7, line 224 - replace 'has been' by 'was'
line 225 - 'mutations' - here, and in several more places later in the text, it would be better not to use this word, which is no longer recommended in the human and medical genetics community.  'Sequence variants', 'variants' or 'pathogenic variants' might be used instead, depending on the context.
line 255 - delete 'such as'
line 265 - replace 'One of' by 'Among' (because fibroblasts are plural)
line 271 (and again later in the text) - replace 'mammal' by 'mammalian'

All done

Page 8, line 308 - 'dependent from' - do the authors mean 'INdependent from' OR 'dependent on' (which have opposite meanings) ?   

We changed with “dependent on”

Page 9, line 345 - replace 'transversally' by 'generally'
line 364 - 'Lambrechts et al.' - where is the numbered reference citation for this?

Corrections made

Page 10, line 395 - replace 'It' by 'There'
line 397 - replace 'N-termini' by 'N-terminus' or 'N-terminal'
line 422 - delete 'The'
line 434 - 'form' should be 'forms'

All done

Page 11, line 440 - italicize the gene names

Done

Page 12, line 502, replace 'lead' by 'led'
line 504 - replace 'which' by 'who'
line 506 - replace 'resulted' by 'was' (same thing for Page 13, line 562)
line 535 - replace 'pathogenesis' by 'pathogenicity'

Words replaced

Page 13, line 544 - 'others' should be 'other'

Corrected

Page 14, line 574 - 'upstream to' should be 'upstream of'

Changed

Page 15, line 631 - replace 'in' by 'into'
line 632 'In this form, it' - do the authors mean PPanSH?  If so, it would be clearer to just say 'PPanSH'  
line 640 - replace 'exceeding' by 'traversing' or 'crossing'
line 665 - 'oxidated' should be 'oxidized'
line 671 - replace 'from the human sign of disease' to 'from the human disease context' (or something similar)
line 673 - replace 'urged' by 'motivated' or 'prompted'

Corrected

Page 683, line 683 - replace 'takes' by 'causes'
line 687 - delete 'the'
line 692 - 'others' should be 'other'
line 700 - 'played a key role in improving the knowledge', replace by 'have played a key role in improving knowledge'

All done

Reviewer 2 Report

The manuscript by Cavestro et al. is an excellent and well-updated review of the distinct pathologies associated to disfunctions of the CoA biosynthetic route.  In each case, authors give a short overview of symptoms and the genetic characterization. Authors highlight the influence of mutations on metabolism—in the context of diverse model organisms tested according to the literature—and their link to signaling pathways when data is on hand. They further assess the different therapeutic approaches currently available or under trial. The text is well-written and easy to read. I consider the manuscript could be published as it is, yet I have some minor comments related to questions that arose during my reading.

1. line 308. I find the term "genetic interactions" ambiguous.  Such interactions result from many physical mechanisms that can be direct or not. I found interesting in that manuscript  the hint to PINK1 aiding PANK2 mitochondrial import, and then, the suggestion of an impact of  PANK2 deficit on p62 acetylation, thereby affecting—as authors point out—to the organelle QC.

2. Line 370.  In addition to PDH, alpha-KDH needs lipoylation for its activation. Related to the dysfunction of these enzymes, is there any report of retrograde signaling in the context of PKAN?

3. When discussing the effectivity of different compounds, authors highlight differences found between tests in vitro and in vivo or between different organisms. I wonder if authors are aware of accurate data on the ADME(T) properties of the different preparations withineach organism in these trials. Maybe it's worth discussing.

Author Response

  1. line 308. I find the term "genetic interactions" ambiguous. Such interactions result from many physical mechanisms that can be direct or not. I found interesting in that manuscript the hint to PINK1 aiding PANK2 mitochondrial import, and then, the suggestion of an impact of  PANK2 deficit on p62 acetylation, thereby affecting—as authors point out—to the organelle QC.

We thank the reviewer for the suggestion. We replaced “genetic interaction” with “functional interaction”. Moreover, we added the following sentence: “In fact, it has been suggested that fbl mRNA follows a PINK1/Parkin-regulated translational mechanism that is localized on the mitochondrial membrane. Moreover, Fbl availability regulates p62/SQSTM1 homolog by acetylation to promote mitophagy, thus affecting mitochondrial quality control”

  1. Line 370. In addition to PDH, alpha-KDH needs lipoylation for its activation. Related to the dysfunction of these enzymes, is there any report of retrograde signaling in the context of PKAN?

To the best of our knowledge, there is no publication reporting a direct link between alpha-KDH and PKAN. However, we added the following sentence: In addition to PDH, other mitochondrial enzymes are modified by lipoylation, among which are α-ketoglutarate dehydrogenase (αKGDH) and branched-chain α-keto acid dehydrogenase (BCKDH). These enzymes are presumably affected as well, as indicated by decreased levels of lipoylated protein levels in yeast, Drosophila, and human PANK2-deficient cells, although their enzymatic activities have never been measured yet.

  1. When discussing the effectivity of different compounds, authors highlight differences found between tests in vitro and in vivo or between different organisms. I wonder if authors are aware of accurate data on the ADME(T) properties of the different preparations within each organism in these trials. Maybe it's worth discussing.

We appreciate the reviewer's comment, but we really think that this issue is out of the scope of this review. We think that the discussion on the ADME(T) properties of all the preparations within each organism is more appropriate for a pharmacological-oriented review.

Reviewer 3 Report

Dear authors,

I read your article with interest and I think it is very useful for those interested in the field of metabolic genetic diseases. In the attached pdf document, you can find my recommendations and comments related to this manuscript.

In addition, I would suggest a small modification of the title: since the mechanisms involved in the discussed pathologies are not known exactly, you could include in the title possible mechanisms or proposed mechanisms.

Also, there is a recent publication related to PKAN disease (Redesigning therapies for pantothenate kinase–associated neurodegeneration", by Munshi et al., 2022) that I did not find among the works cited, although there is a certain similarity in terms of the structure of the manuscript. Please consider it.

In conclusion, after the corrections above-suggested, I recommend publication of this review.

Author Response

I read your article with interest and I think it is very useful for those interested in the field of metabolic genetic diseases. In the attached pdf document, you can find my recommendations and comments related to this manuscript.

We sincerely thank the reviewer for all the comments and suggestions. We attached the pdf document with our point-by-point answers.  

In addition, I would suggest a small modification of the title: since the mechanisms involved in the discussed pathologies are not known exactly, you could include in the title possible mechanisms or proposed mechanisms.

Although we understand the reviewer suggestion, we really think that modification of the title could be counterproductive and confusing. Actually, also the models are “proposed”, as well as the treatments are “proposed” or “possible” translatable to the clinical practice. These are extensively discussed throughout the manuscript. However, the addition of the above adjectives in the title could make it confusing and too long.

Also, there is a recent publication related to PKAN disease (Redesigning therapies for pantothenate kinase–associated neurodegeneration", by Munshi et al., 2022) that I did not find among the works cited, although there is a certain similarity in terms of the structure of the manuscript. Please consider it.

Thanks for the suggestion. We included the citation to the paper. 
